# Effects of Dietary Supplementation of *Lactobacillus acidophilus* on Blood Parameters and Gut Health of Rabbits

**DOI:** 10.3390/ani12243543

**Published:** 2022-12-15

**Authors:** Elena Colombino, Ilaria Biasato, Alberta Michetti, Maria Gabriella Rubino, Irene Franciosa, Marzia Giribaldi, Sara Antoniazzi, Stefania Bergagna, Giulia Paliasso, Ilario Ferrocino, Laura Cavallarin, Laura Gasco, Maria Teresa Capucchio

**Affiliations:** 1Department of Veterinary Sciences, University of Turin, 10095 Grugliasco, TO, Italy; 2Department of Agricultural, Forestry and Food Sciences, University of Turin, 10095 Grugliasco, TO, Italy; 3Institute of Sciences of Food Production, CNR, 10095 Grugliasco, TO, Italy; 4Istituto Zooprofilattico Sperimentale del Piemonte, 10154 Liguria e Valle d’Aosta, TO, Italy

**Keywords:** rabbit, probiotic, *Lactobacillus acidophilus*, gut health, animal health

## Abstract

**Simple Summary:**

Gastrointestinal diseases are one of the most common causes of death in rabbits. Thus, maintaining a proper gut health is fundamental to guarantee adequate growth performance and welfare of the animals. Probiotics (e.g., *Lactobacillus acidophilus*) have been proposed as valuable alternatives to positively modulate gut health. The aim of this study was to evaluate the effects of *Lactobacillus acidophilus* D2/CSL on biochemical parameters, faecal score, cecal pH, gut histomorphometry, microbiota composition and faecal short-chain fatty acids in rabbits. Overall, the dietary inclusion of 1 × 10^9^ cfu/kg feed once a day of *Lactobacillus acidophilus* D2/CSL did not impair rabbit productive performance, blood biochemical parameters, faecal score, gut morphometry, cecal pH, microbiota and short-chain fatty acids concentration. However, it reduced disease incidence and animal death, suggesting that it could improve disease resistance in rabbits.

**Abstract:**

This study aimed to evaluate the effects of *Lactobacillus acidophilus* D2/CSL (L-1 × 10^9^ cfu/kg feed/day) on biochemical parameters, faecal score (FS), cecal pH, gut morphometry, microbiota and cecal short-chain fatty acid (SCFAs) in rabbits. Three zootechnical trials were performed and in each trial 30 rabbits were allotted to two groups; a probiotic group (L) and a control group (C). At slaughter (day 45), samples of blood, duodenum, jejunum, ileum, liver and spleen were collected and submitted to histomorphometric analyses. Blood biochemical analyses, cecal microbiota and SCFAs determination were also performed. In trial 1 and 3, *L. acidophilus* D2/CSL did not affect productive parameters (*p >* 0.05). However, L group of trial 1 showed a lower morbidity and mortality compared to the control. In trial 2, C group showed a higher daily feed intake (*p* = 0.018) and a positive statistical tendency for live weight and average daily gain (*p* = 0.068). On the contrary, albumin was higher and ALFA-1 globulin was lower in the C group compared to L (*p* < 0.05). In all the trials, FS, cecal pH, histomorphometry, microbiota and SCFAs were unaffected. In conclusion, *L. acidophilus* D2/CSL did not impair growth performances, gut and rabbit’s health, reducing morbidity and mortality.

## 1. Introduction

Domestic rabbits (*Oryctolagus cuniculus*) are herbivores, monogastric and hindgut fermenters that rely on cecotrophy to ensure maximum nutrient absorption from their diet [1]. In fact, cecotrophy is a very characteristic habit of this species that aids to complete the digestion of vegetable components and facilitates the assimilation of nutrients synthesized by cecal bacteria, maintaining gut bacterial populations [2]. Due to the unique physiology of their digestive tract, rabbits usually show a fragile balance in their gut function and frequently suffer from enteric disturbances [2,3]. Particularly, enteritis is one of the main causes of death among rabbits due to ensuing diarrhea and subsequent dehydration [4]. Most cases of enteritis are caused by a combination of factors, including feeding on a low fiber diet, debility, and management-related stress along with the presence of one or more potentially pathogenic organisms [4].

In order to prevent these gastrointestinal diseases, it is fundamental to maintain appropriate gut health. Indeed, gut health has been defined as the absence, prevention or avoidance of intestinal disease so that the animal is able to perform its physiological functions in order to withstand exogenous and endogenous stressors [5]. To guarantee a healthy gut, an efficient mucosal barrier function through an adequate gut morphometry and morphology, a stable and diverse microbiota, and an effective intestinal immunity are needed to resist pathogen colonization and to ensure optimal digestion and absorption of nutrients [6].

Because of its critical importance, several studies have focused on the search of valuable feed additives capable of positively modulating gut health. Among them, the use of probiotics seems to be one of the most promising options [7]. Probiotics are defined as direct-fed microorganisms which can modulate the gut microflora through a competitive exclusion process with pathogens [8]. Most of the employed probiotics are lactic acid bacteria (LAB), a diverse group of Gram-positive, nonsporulating, catalase-negative organisms [9,10]. To date, LAB can produce lactic acid as the major metabolic end-product of carbohydrate fermentation and short-chain fatty acids (SCFAs), which reduce the intestinal pH and create a favorable microenvironment for the proliferation of beneficial bacteria [11]. Different probiotics have been already tested in rabbits such as *Lactobacillus acidophilus*, *Lactiplantibacillus plantarum*, *Lactococcus lactis*, *Bacillus subtilis*, *Bacillus licheniformis*, *Bifidobacterium animalis* and *Enterococcus faecium* [3,12,13,14,15,16,17,18,19]. Most of these studies focused on meat rabbits and the influence of probiotics on growth performances, carcass traits and general health. Particularly, Abdelhady et al. [13] registered improved growth performances and a decrease in serum total cholesterol, triglycerides and glucose in rabbits treated with *Bacillus subtilis* and *Bacillus licheniformis*. Moreover, Amber et al. [12] and Bhatt et al. [3] observed a better average daily gain and feed conversion ratio in rabbits administered with *Lactobacillus acidophilus* (0.8 billion CFU/g or 10^7^ cfu/g) or *L. lactis* (10^7^ cfu/g). However, Bhatt et al. [3] did not observe any positive effects on carcass traits and fatty acid profile. Kadja et al. [16] focused on the effect of *Lacticaseibacillus rhamnosus* GG, *Bifidobacterium animalis* subsp. In Lactis BB-12 and *Saccharomyces boulardii* CNCM I-745 on rabbit blood parameters, observing a significant decrease in total cholesterol and triglycerides. as well as a significant increase in total proteins and albumin plasma levels, in treated groups.

Although probiotics seem to positively modulate growth performances and general health in meat rabbits, only a few studies evaluated gut health-related parameters after the administration of different probiotics, obtaining heterogenous results [14,15,19,20]. In fact, Oso et al. did not record any improvement in ileal morphology after the administration of *Pediococcus acidilactis* and *Bacillus cereus* in weaner rabbits. Similarly, gut morphology was also unaffected by the administration of *Lacticaseibacillus casei* in suckling rabbits, even though it reduces the relative abundance of intestinal *Escherichia-Shighella* population [20]. On the contrary, Simonova et al. [14] observed an improvement in intestinal morphology after the administration of fresh culture of *Enterococcus faecium* (5.0 × 108 CFU/animal/day). As far as *Lactobacillus acidophilus* strains are considered, Nwachukwu et al. [19] reported an improvement of the ileal morphometry after its administration. Furthermore, increased Lactobacilli and decreased coliforms, as well as decreased *Listeria monocytogenes* loads, were identified in the gut of weaning [21] and *L-monocytogenes*-challenged weaned rabbits [22], respectively. However, no studies are available on the effects of *Lactobacillus acidophilus* D2/CSL strain on the gut health of domestic rabbits. *Lactobacillus acidophilus* D2/CSL is a probiotic registered in cats, dogs and birds (including poultry) in the European Union, whose beneficial effects have been largely documented [23,24,25,26,27,28,29,30]. Thus, the aim of this study was to evaluate the effects of an oral paste containing *Lactobacillus acidophilus* D2/CSL (1 × 10^9^ cfu/kg feed/day) on biochemical parameters, faecal score, cecal pH, gut morphometry, microbiota composition and SCFAs in domestic rabbits in order to register the new feed additive in EFSA (European Food Safety Agency). Productive parameters were also recorded.

## 2. Materials and Methods

### 2.1. Animals and Diets

The number of animals included in the study was chosen in order to be statistically relevant, according to EFSA Journal Guidance on the assessment of the efficacy of feed additives. In particular, a total of 90 35-day-old healthy commercial hybrid rabbits (45 males and 45 females) were included in the project. In order to register a zootechnical additive, at least three trials in at least two different locations are requested [31]. As a consequence, two double-blinded placebo-controlled trials (trial 1 and trial 2) were performed in the experimental facility of the University of Turin (Italy) and the third one was performed on a rabbit farm located in Cuneo province (Italy).

In each trial, 30 rabbits (15 males and 15 females) were allotted to two groups: a probiotic group, with the administration of an oral paste containing Lactobacillus acidophilus (L) D2/CSL (CECT 4529), at the recommended dosage of 1 *×* 10^9^ cfu/kg feed once a day and a control group (C), receiving a placebo oral paste. The dosage was chosen in accordance with the bibliography [32]. For the oral paste preparation, a high-speed mixer was used (Axomatic, Milan, Italy). The oily raw materials (sunflower oil, soybean oil and malt extract) were mixed and homogenized with glycerol monostearate and lecithin until reaching a temperature of 75 °C and the complete dissolution of all the ingredients. The solid raw materials (Lactobacillus acidophilus and maltodextrins) were premixed in a horizontal mixer (Novinox, Nova Milanese, Italy) for 5 min. The oily phase was cooled to 50 °C under vacuum and then the solid raw materials were added under continuous stirring. Thus, the paste was cooled at room temperature and then transferred to the filling machine for the preparation of the syringes used for the product administration during the study.

After one week of acclimatization, a 2-phase feeding program was applied: a first-period pelleted diet (days 0 to 14) and a second-period pelleted diet (days 15 to 45). Both diets (Mangimi Monge, Torre San Giorgio, CN, Italy) were based on wheat bran, sunflower seed extraction meal, dried beet pulp, alfa-alfa meal, soy husks, sugar cane molasses and soy oil added with a mineral–vitamin premix and coccidiostats to fulfill the nutrient requirements of rabbits (Table 1) [33] Both water and feed were provided ad libitum.

After their arrival at the experimental facilities, all the rabbits were individually weighed (initial live weight, LW) using a precision balance (Sartorius-Signum*^®^*, Bovenden, Germany) and randomly allocated into single cages with individual drinkables and mangers. The facilities were provided with automatic heater and lighting systems, which maintained a constant temperature of 22 °C and set up a lighting schedule of 16 h light and 8 h darkness, respectively.

All the trials lasted 45 days to respect the minimum duration of 42 days required by the EFSA guide for registering additives [31].

### 2.2. Productive Parameters

Health status, clinical illness (morbidity) and mortality were monitored daily throughout each trial. The body condition score (BCS) and faecal score (FS) were recorded per each rabbit. The final live weight (LW) was also recorded at the end of each trial, and the average daily gain (ADG), daily feed intake (DFI), and feed conversion ratio (FCR) calculated as well. The rabbits were carefully managed in order to avoid any potential stress.

The BCS of the animals was assessed using the score proposed by the Pet Food Manufacturer’s Association [34]. The score ranged from 1 to 5 and it was assigned based on a visual and tactile examination of the rabbits.

The faecal score (FS) was assessed by visual observation of the faeces using the score proposed by Weaver et al. [35]. In order to perform this evaluation, faeces were collected under the cage for a 24 h period.

### 2.3. Blood Analysis

At slaughter, blood sample was collected from each rabbit from the jugular vein. An aliquot of 2.5 mL was placed in a serum-separating tube and centrifugated for 5 min at 4000 RPM. The total protein was quantified using the “biuret method” (Bio Group Medical System kit; Bio Group Medical System, Talamello (RN), Italy) and the electrophoretic pattern of the serum was assessed using a semi-automated agarose gel electrophoresis system (Sebia Hydrasys*^®^*, Norcross, GA, USA). Following, electrophoresis, the serum proteins were separated into 5 different types of fractions (protidogram): albumin, α1-Globulins, α2-Globulins, β-Globulins and γ-Globulins. The results were interpreted using the reference values reported by Melillo (2013).

The alanino-aminotransferase (ALT), aspartate-aminotransferase (AST), gamma glutamyl transferase (GGT), alkaline phosphatase (ALP), triglycerides, cholesterol, Na, Cl, P, urea, and creatinine serum concentrations were measured through an automatic analyser (ILab Aries, Instrumental Laboratories, Milano, Italy). Results were interpreted using the reference values reported by Kaneko, Harvey and Bruss [36].

### 2.4. Histomorphometric Investigations

At the end of each trial, all the rabbits were submitted to morphometric and histopathological evaluation. At slaughter, samples of duodenum (after the pylorus), jejunum (middle portion) and ileum (before ileo-caecal junction) were excised and flushed with 0.9% saline to remove all the content. Samples of liver, spleen, kidney and caecum were also collected.

The collected samples were fixed in 10% buffered formalin solution, routinely embedded in paraffin wax blocks, sectioned at 5 μm thickness, mounted on glass slides and stained with Haematoxylin & Eosin (HE). One slide per each intestinal segment was examined by light microscopy and captured with a Nikon DS-Fi1 digital camera (Nikon Corporation, Minato, Tokyo, Japan) coupled to a Zeiss Axiophot microscope (Carl Zeiss, Oberkochen, Germania) using a 2.5× objective lens. NIS-Elements F software was used for image capturing.

Morphometric analysis was performed by Image*^®^*-Pro Plus software (6.0 version, Media Cybernetics, Rockville, Maryland, USA) on 10 well-oriented and intact villi and 10 crypts chosen from each gut segment. The evaluated morphometric indices were as follows: villus height (Vh, from the villus tip to the crypt bottom), crypt depth (Cd, from the crypt bottom to the submucosa) and the villus height to crypt depth ratio (Vh/Cd) [37].

The observed histopathological findings were evaluated in all the organs using a semi-quantitative scoring system as follows: absent (score = 0), mild (score = 1), moderate (score = 2) and severe (score = 3). Gut histopathological findings were separately assessed for mucosa (inflammatory infiltrates) and submucosa (inflammatory infiltrates and Gut-Associated Lymphoid Tissue [GALT] activation) for each segment. The total score of each gut segment was obtained by adding up the mucosa and submucosa scores. All the slides were blindly assessed by three independent observers and the discordant cases were reviewed, using a multi-head microscope, until a unanimous consensus was reached.

### 2.5. Cecal pH and Microbiota

At slaughter, caecal pH was measured in duplicate in the caecal appendix of 8 rabbit/treatment using a Crison portable pH-meter (Crison Instruments, S.A., Alella, Spain) fitted with a spear-type electrode and an automatic temperature compensation probe. After pH measurement, the caecal content was collected into sterile plastic tubes and frozen at −20 °C for microbiota analyses. A metataxonomic approach was applied to analyse the total DNA extracted from cecal samples of rabbits of L and C groups in order to highlight any differences in microbiota composition. The 16S rRNA gene (V3-V4 regions) was amplified using primers and procedures defined by Klindworth et al. [38]. The PCR products were purified, tagged and pooled following the Illumina guidelines. Illumina MiSeq platform with V2 chemistry was used to generated 250-bp paired-end reads and the raw files obtained (*.fastq*) were elaborated by QIIME 2 v. 2022.8 software [39]. Cutapter software was used to remove primer sequences, and DADA2 algorithms [40] was used to denoise the obtained reads by using the q2-dada2 plugin in QIIME 2. Taxonomy classification was performed against the SILVA database using the QIIME2 qiime feature-classifier. The Amplicon Sequence Variances (ASVs) with less than five read counts in at least two samples were excluded to increase the confidence of sequence reads. ASV table displayed the lowest taxonomic resolution; when the genus was not reached, family or class was displayed. The BLAST tool was used to confirm the taxonomic assignments.

### 2.6. Cecal Short-Chain Fatty Acids

Following slaughtering, ceca content was collected from 8 rabbits/treatment and immediately frozen and stored at −20 °C until SCFAs analysis. SCFAs quantification was carried out according to the methods described by Guantario et al. [41]. Briefly, samples (200 mg) were suspended in 250 μL of 0.1 N H_2_SO_4_ solution and centrifuged at 15,000× *g* for 10 min at 4 °C. The supernatant was transferred in a glass vial. Analyses were performed on a HPLC (High Performance Liquid Cromatography) Ultimate 3000 Thermo Fisher with autosampler equipped with a 300 × 7.8 mm Aminex HPX-87H and a guard-column. Injected samples (30 μL) were isocratically separated in 0.005 N H_2_SO_4_, at a flow rate of 0.6 mL/min and column temperature 41 °C. SCFAs were detected at 210 nm, using an external standard curve VFAs were detected by UV light at 210 nm and identified using an external standard curve (4.95–148.5 mg/100 mL succinic acid; 9–270 mg/100 mL lactic acid; 10.5–314.4 mg/100 mL acetic acid; 9.85–285.5 mg/100 mL propionic acid; 9.5–285.1 mg/100 mL isobutyric acid) created using standards dissolved in 0.1N H_2_SO_4_. Total VFAs were expressed as mg/100 mL.

### 2.7. Statistical Analysis

GraphPad Prism*^®^* software version 8.0 and R studio software version 4.0.4 (R Foundation for Statistical Computing, Vienna, Austria; http://www.r-project.org [accessed on 5 September 2022]) was used to perform a statistical analysis.

Individual rabbits were considered as experimental units to analyse all the parameters. The Shapiro–Wilk test was used to test the normality of the data distribution before statistical analyses. Data were described by mean and standard deviation (SD) or median and interquartile range (IR) depending on data distribution. Bivariate analysis was performed by Student’s t and Mann–Whitney U tests to compare the growth performance, the BCS, the faecal score, the aaecal pH, the blood biochemical parameters, gut morphology and organs histopathology between the two groups. *p* values ≤ 0.05 were considered statistically significant.

Regarding microbiota analysis, alpha and beta diversity indexes were calculated through the diversity script of QIIME2. Differences between alpha and beta diversity parameters were calculated with Kruskal–Wallis and ANOSIM statistical tests respectively, and ASVs frequency at the lowest taxonomic resolutions (genus or family) were analyzed by non-parametric Kruskal–Wallis test in R environment.

## 3. Results

### 3.1. Productive Parameters

In trial 1, growth performances (LW, DFI, ADG, FCR) did not show any significant differences (*p* > 0.05) between the control and the probiotic rabbits (Table 2). However, the BCS showed a higher mean value in the L than in the C group (*p* = 0.038).

In trial 2, statistically positive trends were observed for LW and ADG (*p* = 0.068), being greater in the C group. Furthermore, DFI was significantly higher in C than in the L group (*p* = 0.018) (Table 2).

In trial 3, the growth performance (LW, DFI, ADG, FCR) did not show any significant differences between the control and the probiotic rabbits, as well as BCS and FS (*p* > 0.05, Table 2).

In trial 1, scattered diseases and deaths were recorded. In particular, two rabbits of the control group showed enteritis (20%); one died, while the other spontaneously recovered. One rabbit in the control group also showed multiple cutaneous abscesses (6.66%). Regarding the probiotic group, one animal suddenly died without showing any clinical symptoms (6.66%). On the contrary, in the second and third trials, all the rabbits remained healthy.

### 3.2. Biochemical Parameters

In all the trials, biochemical parameters did not show any significant differences between the control and the probiotic rabbits (*p* > 0.05; Table 3). In trial 1 and 2, dietary Lactobacillus acidophilus D2/CSL supplementation did not significantly affect the protogram (*p* > 0.05; Table 4). On the contrary, in trial 3, albumin (%) was higher and alpha globulin (g/dL) was lower in the C group compared to the L group (*p* = 0.013 and *p* = 0.020, respectively).

### 3.3. Histomorphometric Investigations

Data regarding morphometric measurements are reported in Table 5. No statistically significant differences were recorded for Vh, Cd and Vh/Cd in duodenum, jejunum and ileum between the control and treated groups (*p* > 0.05) in the three trials.

Regardless of the Lactobacillus acidophilus D2/CSL supplementation, Vh and Vh/Cd showed a proximodistal decreasing gradient from duodenum/jejunum to ileum.

Histopathological alterations developed in all the organs for all the dietary treatments in the three trials (Table 6). In particular, in liver, from absent to moderate multifocal lymphoplasmacytic inflammatory infiltrates were observed with no sign of vacuolar degeneration. In spleen, from absent to moderate multifocal white pulp hyperplasia was detected while duodenum, jejunum, ileum and caeca showed mild to moderate lymphoplasmacytic infiltrates with scattered eosinophils and lymphoid tissue hyperplasia. However, Lactobacillus acidophilus D2/CSL supplementation did not affect the severity of the observed histopathological alterations (*p* > 0.05).

### 3.4. Caecal pH and Microbiota

In all the three trials, the caecal pH was not significantly affected by Lactobacillus acidophilus D2/CSL supplementation (trial 1—C:7.17 [0.08] and L:7.09 [0.05]; trial 2—C:7.12 [0.10] and L: 7.11 [0.07]; trial 3—C: 7.26 [0.35] and L: 7.24 [0.30]).

The Kruskal–Wallis test was used to assessed differences in alpha diversity value (Figure 1A) but the dietary treatment was not statistically significant. The Bray Curtis distance matrix was used to performed PCoA (Figure 1B) and ANOSIM test as a function of the dietary treatment. No significant differences were observed as a function of the probiotic dietary treatment.

In the three trials, the microbiota showed the same metataxonomic composition (Figure 1C), although some significant variations were recorded (*p* < 0.05). In all the rabbits, Clostridia was the most abundant class, Eubacteriaceae, Lachnospiraceae and Muribaculaceae were the most abundant families while Akkermansia, Eubacterium, and Ruminococcus were the most abundant genera in the dataset (Figure 1C).

### 3.5. Caecal Short-Chain Fatty Acids

Table 7 summarized the results of caecal SCFAs. In all the three trials, dietary supplementation of Lactobacillus acidophilus D2/CSL did not affect acetic acid, isobutyric acid and propionic acid concentration (*p* > 0.05). Furthermore, in trial 1 and 2, non-significant differences were recorded for lactic and succinic acids between the two groups (*p* > 0.05). In trial 3, lactic acid was undetermined, while succinic acid displayed higher values in supplemented rabbits than non-supplemented ones (*p* < 0.001).

## 4. Discussion

This study aimed to evaluate the effects of *Lactobacillus acidophilus* D2/CSL on blood parameters and gut health (faecal score, caecal pH, morphometry, microbiota composition, and caecal SCFAs) of rabbits. Productive performances were also evaluated, even if the number of the involved animals was limited and these parameters were not requested by the EFSA Journal Guidance on the assessment of the efficacy of feed additives. To date, in trials 1 and 3, D2/CLS *L. acidophilus* integration did not affect the productive performances (*p* > 0.05). However, the treated group of trial 1 showed a higher BCS compared to the control group (*p* < 0.05). In trial 2, the control group showed a higher DFI (*p* = 0.018) and a positive statistical tendency for LW and ADG (*p* = 0.068) compared to the treated rabbits. On the contrary, albumin was higher and ALFA-1 globulin was lower in the C group compared to treated rabbits (*p* < 0.05). In all the trials, biochemical parameters, faecal score, caecal pH, histomorphometry, microbiota and SCFAs were unaffected by *L. acidophilus* D2/CSL administration.

To date, the lack of effects observed on productive performances in trials 1 and 3 is in contrast with the available literature. In fact, the majority of the studies reported improved growth performances after the administration of different probiotics [3,12,21]. In particular, Amber et al. [12] reported improved ADG and FCR in rabbits receiving *Lactobacillus acidophilus* (4 × 10^5^ cfu/g diet) when compared to the control group. Similarly, Lam Phuoc et al. [21] reported a greater LW in rabbits fed diets supplemented with *Lactobacillus acidophilus* (1 × 10^7^ cfu/g) or a mixture of *Lactobacillus acidophilus* and *Bacillus subtilis* between day 40 and 70 of the trial, probably due to the decrease in the intestinal coliform population. Furthermore, Bhatt et al. [3] reported an improved LW in treated rabbits after the administration of *Lactobacillus acidophilus* (107 cfu/g concentrate) compared to the control group (24.5 g/day vs. 22.5 g/day, respectively).

The discrepancy between the results recorded for growth performances in the present studies and the available literature could be related to the different dosage of the administered probiotic and to the different trial duration. In fact, all the above-mentioned studies administered a higher dosage (4 × 10^5^ cfu/g diet; 10^7^ cfu/g concentrate; 1 × 10^7^ cfu/g) for longer periods (49, 63 and 70 days) [3,12,21]

However, the higher DFI recorded for the control group in trial 2 could explain the statistical tendency observed also for LW and ADG in the same group. Indeed, the control group showed a numerically higher FCR compared to the treated groups and it is reasonable to hypothesize that the higher growth rate could be attributed to the higher feed intake. As a consequence, treated rabbits seem to have a better FCR (3.57 vs. 3.48 in control and treated groups, respectively). Although the majority of the studies reported that DFI is not influenced by probiotic administration [3,12], effects of probiotics on FCR are still controversial. The findings of the present studies are in agreement with El-Katcha et al. [17] who observed unaffected FCR in rabbits administered with *Lactobacillus* strains. On the contrary, Amber et al. [12] observed a significant reduction of FCR in rabbits administered with *Lactobacillus acidophilus* while Abdel-Aziz et al. [42], Bhatt et al. [3] and Lam Phuoc et al. [21] revealed a worsening of the FCR in rabbits fed diets containing probiotics.

Moreover, in the trial 1, dietary *Lactobacillus acidophilus* D2/CSL supplementation determined an increase in the rabbit BCS. This finding could be related to the reduction of disease occurrence in the treated group compared to control. In fact, the control group showed enteritis and cutaneous abscesses, which probably determined a reduction in the feed consumption and, in turn, a worsening of the nutritional status of the animals. Furthermore, the improvement in the immune status of rabbits can be associated with enhanced growth performance [43], and BCS also seems to be correlated with the body weight [44]. This hypothesis is confirmed by the lack of effect of *L. acidophilus* D2/CSL supplementation on the BCS of the rabbits of trial 2 and 3, in which no sign of diseases was recorded neither in the treated nor in the control rabbits, suggesting a good health status in both groups. These findings are partially in agreement with Lam Phuoc et al. [21], who showed a significant decrease in morbidity in rabbits fed diets containing *Lactobacillus acidophilus*, as well as no mortality. Similarly, Abdel-Azeem et al. [45] reported a significant reduction in mortality in the rabbits administered with probiotic.

Regarding FS, it was unaffected by dietary *L. acidophilus* D2/CSL supplementation in all the trials. These findings are in contrast with the results of Lam Phuoc et al. [21], who identified lower faecal scores in the probiotic-fed rabbits compared to the control rabbits. A previous study from the same authors also revealed a lower faecal consistency index in rabbits administered with *Lactobacillus acidophilus* probably due to an increase in SCFAs, which provide a powerful driving force for the movement of water and sodium out of the colonic lumen, leading to reduced moisture content in the faeces and therefore a lower faecal score when compared to the control group [21].

As already mentioned above, the different findings observed in the present trials could be related to the different strain and dosage of the administered probiotic (1 × 10^9^ cfu/kg feed vs. 1 × 10^7^ cfu/g feed).

In both trials 1 and 2, a clinical chemistry analysis and electrophoresis for serum proteins were not affected by the administration of *Lactobacillus acidophilus* D2/CSL and all the parameters fell within the physiological ranges [36,46]. These results are partially in agreement with El-Adawy et al. [47], who reported non-significant effects on plasma total proteins, GOT, GPT, creatine and urea concentration after dietary *Lactobacillus acidophilus* supplementation. Similarly, Abdel-Azeem et al. [45] identified no significant differences for the creatinine and the urea concentration between the control- and the probiotic ZAD^®^- fed rabbits, with the parameters also falling within the physiological ranges. However, ALT and AST significantly decreased and A/G ratio significantly increased in rabbits that received the probiotic supplementation when compared to the control animals [45].

On the contrary, in trial 3, albumin (%) was higher in the C group compared to the treated rabbits. These results are in contrast with the previous study of Kadja et al. [16], who found higher albumin levels in the probiotic groups *(Lactobacillus rhamnosus*, *Bifidobacterium animalis* or *Saccharomyces boulardii*) compared with the control group. Also, in their study using *Lactobacillus rhamnosus*, Simonova et al. [14] found slightly higher total protein levels. Furthermore, an increase in the concentration of proteins and albumin was reported in rabbits supplemented with 10^6^ cfu/g of *Lactobacillus planetarium* for 8 weeks [48]. However, in trial 3, ALFA-1 globulin was higher in the treated group (*p* < 0.05). Despite the statistical significance, in trial 3, the albumin levels were numerically similar in both groups and the treated group presented a slightly higher blood total proteins and higher ALFA-1 globulins fraction, being partially in accordance with literature. This increase in plasma ALFA-1 globulin fraction could be a consequence of the beneficial effects of probiotics on protein metabolism in the gut and the increase in the level of globulins could indicate a possible improvement in the immunity of rabbits [16].

As far as histomorphometry is concerned, no significant differences were observed between control and probiotic groups (*p* > 0.05) in the present trials.

In literature, no studies are available on gut histology and morphometry in rabbits fed with *Lactobacillus acidophilus* D2/CSL and the results obtained with other probiotics supplementation are controversial. Seyidoglu et al. [49] found similar results after administrating *Saccharomyces cerevisiae* to rabbits. Furthermore, Oso et al. [15] did not record any improvement in ileal morphology after the administration of *Prediococcus acidilactis* and *Bacillus cereus* in weaner rabbits. Similarly, gut morphology was also unaffected by the administration of *Lactobacillus casei* in suckling rabbits [20]. On the contrary, Simonová et al. [14] observed greater jejunum villus height, surface area and Vh/Cd along with reduced Cd in rabbits receiving *Enterococcus faecium* CCM7420 supplementation. Moreover, Nwachukwu et al. [19] reported an improvement of the ileal Vh, Cd and Vh/Cd after the administration of *Lactobacillus acidophilus*.

Regardless of diet, the present study confirmed that the morphometric indices followed a proximodistal decreasing gradient from the duodenum/jejunum to the ileum. This is related to the intestinal absorption processes, which evolved differently depending on the considered segment. Indeed, the duodenum is the intestinal tract with the fastest cell renewal, and it is also the first segment to receive physical, chemical and hormonal stimuli provoked by the presence of the diet in the lumen [1]. Moreover, *Lactobacillus acidophilus* D2/CSL did not affect the histopathological features of the rabbits of the present trials, thus suggesting no negative influence on animal health.

Regarding microbiota composition, it was not influenced by diet in all the trials (*p* > 0.05). In all the rabbits, Clostridia was the most abundant class, *Eubacteriaceae*, *Lachnospiraceae* and *Muribaculaceae* were the most abundant family while *Akkermansia*, *Eubacterium*, and *Ruminococcus* were the most abundant genera. These findings are in accordance with Bauerl et al. [2], Combes et al. [50] and Cotozzolo et al. [51], who reported Clostridia, *Lachnospiraceae*, *Ruminococcus*, *Eubacterium* and *Akkermansia* as the most prevalent class, family and genera in the rabbit caeca. Particularly, Clostridia class encompasses a great number of bacterial genera included in the Firmicutes phylum that physiologically inhabited the rabbit caeca [52]. It includes the *Clostridium* genera, which have been reported in a higher percentage in caecal microbiota of rabbits affected by epizootic rabbit enteropathy than in healthy animals. However, not all the *Clostridium* species are pathogenic and most of them are cellulose-degrading symbiotic microorganisms that help the host in the digestion of plant materials [52]. Little is known on the *Eubacteriaceae* family role in rabbits, but *Eubacterium* genus seems to be able to produce butyrate, which plays a critical role in energy homeostasis, colonic motility, immunomodulation and suppression of inflammation in the gut [53]. Moreover, both *Lachnospiraceae* family and *Ruminococcus* genus are considered important indexes of intestinal health [51]. In fact, they play an important role in the degradation of vegetable feed components and the production of SCFAs [2]. Regarding *Akkermansia* genus, it is part of the Verrucomicrobia phylum and it has been reported to positively regulate the production of antioxidant metabolites, protect the self-healing of the intestinal mucosal protective layer, and enhance the response to inflammatory reaction damage [6] The most well-known species in this genus is *Akkermansia muciniphila*, a mucin-degrading bacterium that has been demonstrated to be beneficial in rabbits as its ability to break down mucin is especially important during caecotrophy for optimal nutrient extraction [4]. Finally, the presence of *Muribaculaceae* family has not been reported previously in rabbit caeca but in mink gut it seems to be able to degrade complex carbohydrates [54].

The lack of effects observed on microbiota composition after the administration of *L. acidophilus* D2/CSL is in contrast with the results of Amber et al. [12], in which the addition of *L. acidophilus* increased the number of cellulolytic bacteria and reduced the urolithic ones in rabbit caeca. Moreover, Shen et al. [20], Lam Phuoc et al. [21] and Abdel-Azeem et al. [45] reported an increment in lactobacilli count and a decrease in coliforms after probiotic administration. However, the results of the present trial are in accordance with Beshara et al. [55], who reported a lack of variation in LAB count in the probiotic-fed rabbits. The heterogenicity of results seems to support that the effect of probiotics depended on the strain, dose, and time of administration [56].

Finally, faecal SCFA were unaffected by *L. acidophilus* D2/CSL administration in all the three trials, with the only exception of the succinic acid in trial 3. SCFAs are important metabolites derived from gut microbial fermentation of complex carbohydrates, which have been proven to play a key role in intestinal health and energy homeostasis regulation [57]. Particularly, they represent an important energy source for the enterocytes and they showed strong anti-inflammatory effects that positively affect the body weight gain of animals [57]. The results of the present study are partially in accordance with Oso et al. [15], who observed unaffected acetic, propionic and butyric acid in rabbits receiving dietary probiotic inclusion (*Prediococccus acidilactis*, *Bacillus cereus*). The lack of variations herein reported can be due the lack of variations in the SCFAs-producing bacteria in the caecal microbiota of control and treated groups or to a major effect of *L. acidophilus* D2/CSL in the upper part of the gastrointestinal tract not detectable at the caecal level. In fact, *Lactobacillus* do not normally inhabit the caeca of rabbits and it seems to poorly adhere to epithelial cells; possibly reducing its efficacy on caecal microbiota and SCFAs production [58].

## 5. Conclusions

In conclusion, in the present study *Lactobacillus acidophilus* D2/CSL administration at the dosage of 1 × 10^9^ cfu/kg feed determined no overall significant effects on the biochemical parameters, the gut histomorphology, the caecal pH, microbiota and SCFAs of growing rabbits. On the contrary, the trial 1-rabbits supplemented with the probiotic showed improved BCS as a consequence of the absence of disease development, thus representing a positive outcome as well as in trial 3 the increase of ALFA-1 globulin in the treated group could indicate a possible improvement in the immunity of rabbits.

This improvement recorded in the rabbits BCS and immune status could be useful both in pet and commercial rabbits as it can help in reducing the risk of disease development and in improving the rabbit’s well-being.

## Figures and Tables

**Figure 1 animals-12-03543-f001:**
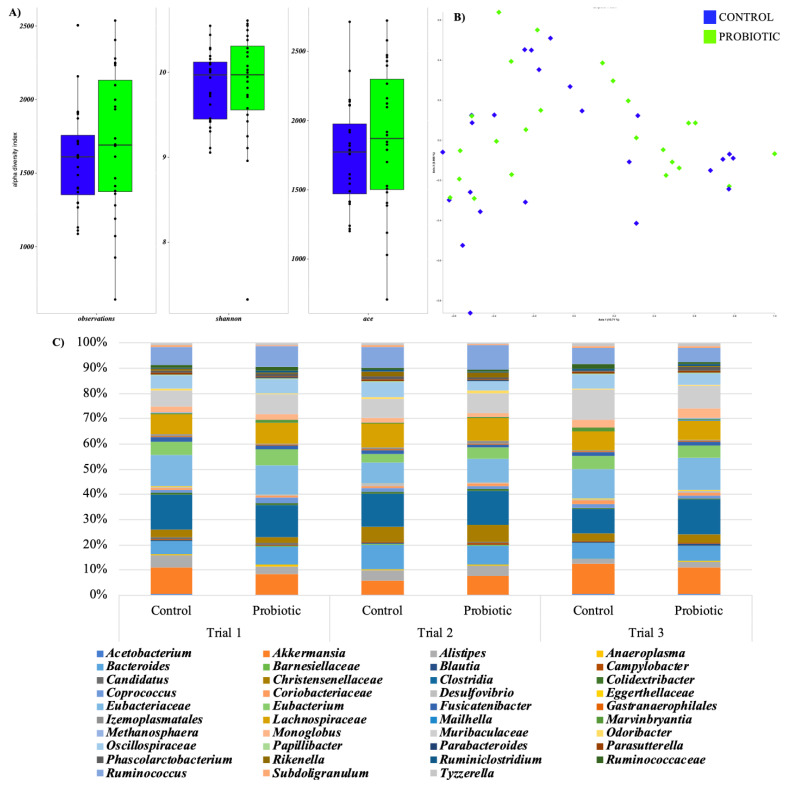
(**A**) Boxplots to describe α-diversity measures of the caecal microbiota of rabbits fed with probiotic (green bars) or control diet (blue bars). Individual points and brackets represent the richness estimate and the theoretical standard error range, respectively. (**B**) Principal Coordinate analysis (PCoA) based on Bray Curtis distance matrix as a function of the dietary treatment. (**C**) Metataxonomic composition of caecal microbiota in probiotic (L) and control (C) groups from trial 1, 2 and 3 at the lowest taxonomic resolution (genus or family).

**Table 1 animals-12-03543-t001:** Analytical components of feeds reported on the labelling administered to rabbits.

Analytical Components (%)	Dietary Treatments	
	First Period (0–14 Day)	Second Period (15–45 Day)
CP	15.50	16.30
Fat	2.70	2.50
CF	17.50	15.80
Ash	7.60	7.80
Ca	1.10	1.10
P	0.50	0.60
Na	0.20	0.30

CP = crude protein, CF = crude fiber, Ca = calcium, P = phosphorus, Na = sodium.

**Table 2 animals-12-03543-t002:** Productive parameters of trial 1, 2 and 3.

	Trial 1			Trial 2			Trial 3		
	C	L	*p*-Value	C	L	*p*-Value	C	L	*p*-Value
**LW (g)**, *mean (SD)*	3131.00 (97.53)	3242.00 (106.40)	0.448	3430.10 (258.76)	3188.90 (419.52)	0.068	3459.4 (375.89)	3535.9 (299.06)	0.542
**DFI (g)**, *mean (SD)*	155.20 (6.95)	163.20 (7.15)	0.429	184.27 ^a^ (5.33)	160.59 ^b^ (7.80)	0.018	160.24 (20.61)	162.08 (19.39)	0.807
**ADG (g)**, *mean (SD)*	45.40 (2.44)	48.01 (2.41)	0.455	51.77 (1.67)	46.30 (2.34)	0.068	47.73 (6.44)	49.44 (5.27)	0.444
**FCR (g)**, *mean (SD)*	3.47 (0.12)	3.43 (0.07)	0.745	3.57 (0.05)	3.48 (0.06)	0.287	3.36 (0.15)	3.28 (0.17)	0.164
**BCS (1–5)**	2.91 ^a^ (0.04)	3.00 ^b^ (0.00)	0.038	3.07 (0.04)	3.02 (0.06)	0.225	3.20 (0.08)	3.14 (0.07)	1.000
**FS (0–4)**	3.76 (0.10)	3.96 (0.01)	0.131	3.98 (0.02)	3.98 (0.01)	0.633	3.92 (0.04)	3.97 (0.02)	0.594

ADG = average daily gain; BCS = body condition score; C: control; DFI = daily feed intake; FCR = feed conversion ratio; FS = faecal score; L: Lactobacillus; LW = live weight. Means with superscript letters denote significant differences (*p* < 0.05).

**Table 3 animals-12-03543-t003:** Clinical chemistry analysis between control and probiotic group of trial 1, 2 and 3.

Trial 1				Trial 2			Trial 3		
	C	L	*p*-Value	C	L	*p*-Value	C	L	*p*-Value
**ALT (U/L)**, *mean (SD)*	54.50 (6.85)	45.08 (4.16)	0.253	51.08 (3.86)	49.17 (4.76)	0.758	45.58 (5.56)	48.58 (2.55)	0.164
**AST (U/L)**, *mean (SD)*	33.25 (3.59)	33.92 (3.83)	0.900	27.58 (3.04)	33.75 (3.56)	0.201	38.33 (2.46)	38.33 (2.22)	>0.999
**COL (mg/dL)**, *mean (SD)*	48. 83 (4.15)	48.33 (3.34)	0.926	43.00 (3.34)	47.00 (3.72)	0.432	48.25 (3.76)	49.33 (4.76)	0.860
**CRE (mg/dL)**, *mean (SD)*	0.84 (0.05)	0.94 (0.05)	0.162	0.87 (0.04)	0.87 (0.05)	0.909	1.06 (0.04)	0.95 (0.04)	0.102
**GGT (U/L)**, *mean (SD)*	9.00 (1.23)	10.92 (2.73)	0.529	7.17 (0.66)	8.42 (2.46)	0.629	9.00 (1.00)	7.58 (1.49)	0.438
**PRTOT (g/dL)**, *mean (SD)*	6.07 (0.15)	6.05 (0.11)	0.893	6.20 (0.09)	6.15 (0.11)	0.734	6.32 (0.16)	6.33 (0.13)	0.969
**TRIGL (mg/dL)**, *mean (SD)*	56.67 (4.45)	54.42 (3.33)	0.689	52.00 (4.03)	51.50 (2.49)	0.917	43.33 (1.74)	40.75 (1.75)	0.305
**UREA (mg/dL)**, *mean (SD)*	18.17 (0.95)	18.92 (0.76)	0.545	19.17 (1.21)	17.67 (1.12)	0.372	53.17 (2.70)	50.75 (2.63)	0.528
**Cl (mmol/L)**, *mean (SD)*	109.30 (2.67)	109.70 (1.96)	0.887	99.83 (3.49)	98.74 (3.00)	0.816	95.81 (1.03)	95.83 (0.88)	0.985
**K (mmol/L)**, *mean (SD)*	9.95 (0.45)	10.09 (0.53)	0.847	9.13 (0.42)	14.05 (5.02)	0.339	11.70 (0.55)	11.22 (0.42)	0.500
**Na (mmol/L)**, *mean (SD)*	161.20 (3.35)	161.80 (2.64)	0.888	150.3 (4.22)	148.2 (3.56)	0.720	138.7 (1.22)	137.9 (1.27)	0.678

C = control group; L = probiotic group with Lactobacillus acidophilus D2/CSS; ALT = alanine aminotranpherase; AST = aspartate aminotranpherase; COL = cholesterol; CRE = creatinine; GGT = **γ** glutamyl transpherase; PRTOT = total protein; TRIGL = triglycerides; Cl = chlorine; K = potassium; Na = sodium.

**Table 4 animals-12-03543-t004:** Electrophoresis of serum proteins between control and probiotic group of trials 1,2 and 3.

	Trial 1			Trial 2			Trial 3		
	C	L	*p*-Value	C	L	*P*-Value	C	L	*p*-Value
**ALB (%)**, *mean (SD)*	60.90 (0.90)	60.45 (0.81)	0.714	60.45 (0.81)	59.58 (1.35)	0.589	68.88 (0.78)	67.28 (0.57)	0.013
**ALB (g/dL)**, *mean (SD)*	3.70 (0.10)	3.66 (0.07)	0.732	3.66 (0.07)	3.66 (0.08)	0.976	4.35 (0.11)	4.26 (0.09)	0.522
**ALFA-1 (%)**, *mean (SD)*	7.73 (0.38)	7.46 (0.20)	0.529	7.46 (0.20)	7.25 (0.27)	0.548	7.33 (0.22)	7.97 (0.23)	0.060
**ALFA-1 (g/dL)**, *mean (SD)*	0.47 (0.02)	0.45 (0.01)	0.527	0.45 (0.01)	0.44 (0.01)	0.777	0.46 (0.01)	0.50 (0.01)	0.020
**ALFA-2 (%)**, *mean (SD)*	6.96 (0.15)	7.35 (0.23)	0.168	7.35 (0.23)	8.25 (0.66)	0.210	7.07 (0.20)	7.16 (0.23)	0.765
**ALFA-2 (g/dL)**, *mean (SD)*	0.42 (0.01)	0.44 (0.01)	0.191	0.44 (0.01)	0.51 (0.04)	0.165	0.45 (0.02)	0.45 (0.02)	0.798
**BETA (%)**, *mean (SD)*	13.08 (0.91)	12.89 (0.58)	0.867	12.89 (0.58)	11.51 (0.75)	0.157	8.92 (0.19)	8.88 (0.23)	0.890
**BETA (g/dL)**, *mean (SD)*	0.79 (0.06)	0.78 (0.04)	0.852	0.78 (0.04)	0.71 (0.05)	0.288	0.57 (0.02)	0.56 (0.01)	0.787
**GAMMA (%)**, *mean (SD)*	11.33 (0.67)	11.85 (0.75)	0.612	11.85 (0.75)	13.41 (0.99)	0.223	7.79 (0.59)	8.70 (0.68)	0.327
**GAMMA (g/dL)**, *mean (SD)*	0.69 (0.04)	0.72 (0.05)	0.680	0.72 (0.05)	0.83 (0.07)	0.218	0.49 (0.04)	0.55 (0.05)	0.354
**A/G**, *mean (SD)*	1.57 (0.05)	1.54 (0.05)	0.680	1.54 (0.05)	1.50 (0.08)	0.705	6.32 (0.16)	6.33 (0.14)	0.969

C = control group; L = probiotic group with Lactobacillus acidophilus D2/CSL; ALB = albumin; A/G = albumins/globulins ratio.

**Table 5 animals-12-03543-t005:** Morphometric evaluation of the small intestine between control and probiotic group of trial 1, 2 and 3.

Trial 1				Trial 2			Trial 3		
	C	L	*p*-Value	C	L	*p*-Value	C	L	*p*-Value
**Duodenum**									
Vh, *mean (SD)*	0.84 (0.16)	0.84 (0.25)	0.977	0.72 (0.25)	0.81 (0.27)	0.291	0.95 (0.12)	0.94 (0.15)	0.921
Cd, *mean (SD)*	0.05 (0.01)	0.04 (0.01)	0.339	0.04 (0.01)	0.04 (0.01)	0.629	0.05 (0.01)	0.05 (0.01)	0.599
Vh/Cd, *mean (SD)*	16.04 (6.07)	18.75 (5.15)	0.409	15.53 (6.55)	18.70 (6.15)	0.233	18.46 (3.22)	18.89 (4.33)	0.781
**Jejunum**									
Vh, *mean (SD)*	0.76 (0.17)	0.82 (0.19)	0.498	0.78 (0.16)	0.81 (0.13)	0.665	0.93 (0.12)	0.97 (0.12)	0.517
Cd, *mean (SD)*	0.05 (0.01)	0.04 (0.01)	0.400	0.05 (0.01)	0.05 (0.01)	0.986	0.05 (0.01)	0.05 (0.01)	0.589
Vh/Cd, *mean (SD)*	16.86 (4.49)	18.81 (4.75)	0.313	16.07 (2.73)	17.09 (4.14)	0.480	18.19 (2.70)	19.21 (3.05)	0.396
**Ileum**									
Vh, *mean (SD)*	0.61 (0.20)	0.58 (0.16)	0.752	0.63 (0.12)	0.60 (0.16)	0.562	0.54 (0.08)	0.52 (0.07)	0.537
Cd, *mean (SD)*	0.05 (0.01)	0.04 (0.01)	0.434	0.04 (0.01)	0.04 (0.01)	0.843	0.05 (0.01)	0.04 (0.01)	0.029
Vh/Cd, *mean (SD)*	12.88 (4.99)	12.50 (3.29)	0.826	14.29 (3.77)	13.23 (4.06)	0.516	11.27 (2.69)	12.17 (2.84)	0.434

C = control group without any treatment; L = probiotic group with Lactobacillus acidophilus D2/CSL; Vh = villus height; Cd = crypt depth; SD = standard deviation; IR = interquartile range.

**Table 6 animals-12-03543-t006:** Histopathological alterations of the main organs between control and probiotic group of trial 1, 2 and 3.

Trial 1				Trial 2			Trial 3		
	C	L	*p*-Value	C	L	*p*-Value	C	L	*p*-Value
Liver, *median (IR)*	0.25 (0.00–1.00)	0.00 (0.00)	0.651	0.58 (0.00–1.00)	0.62 (0.00–1.00)	0.999	0.50 (0.50–1.00)	0.50 (0.12–0.87)	0.286
Spleen, *mean (SD)*	0.22 (0.51)	0.33 (0.49)	0.640	0.00 (0.00)	0.18 (0.60)	0.640	0.00 (0.00)	0.00 (0.00)	>0.999
Duodenum, *median (IR)*	0.50 (0.00–0.50)	0.50 (0.00–0.75)	0.477	0.50 (0.00–0.50)	0.50 (0.00–1.00)	0.115	0.50 (0.50–1.00)	0.50 (0.00–0.87)	0.882
Jejunum, *median (IR)*	0.50 (0.00–1.00)	0.50 (0.00–0.50)	0.352	0.50 (0.00–0.87)	0.50 (0.00–0.50)	0.710	0.50 (0.12–0.50)	0.50 (0.00–0.50)	0.400
Ileum, *median (IR)*	0.25 (0.00–0.50)	0.50 (0.00–1.00)	0.379	0.25 (0.00–0.50)	0.50 (0.00–1.00)	0.246	0.50 (0.00–0.50)	0.50 (0.00–0.50)	0.689
Caecum, *median (IR)*	0.45 (0.49)	0.37 (0.48)	0.840	0.50 (0.00–1.00)	0.25 (0.00–1.00)	0.576	0.00 (0.00–0.50)	0.50 (0.00–0.87)	0.146

C = control group without any treatment; L = probiotic group with Lactobacillus acidophilus D2/CSL; SD = standard deviation; IR = interquartile range.

**Table 7 animals-12-03543-t007:** Faecal short-chain fatty acids detected in trials 1, 2 and 3.

	Trial 1			Trial 2			Trial 3		
	C	L	*p*-Value	C	L	*p*-Value	C	L	*p*-Value
Acetic acid, *mean (SD)*	139.50 (135.8)	148.90 (73.05)	0.346	80.05 (39.90)	78.91 (37.05)	0.944	84.82 (33.18)	128.70 (93.94)	0.198
Lactic acid, *mean (SD)*	111.30 (156.1)	121.00 (72.12)	0.167	142.00 (140.50)	91.90 (50.60)	0.315	n.d	n.d	n.d
Succinic acid, *mean (SD)*	81.78 (42.62)	112.00 (66.11)	0.258	58.21 (79.00)	54.44 (20.25)	0.143	38.67 (50.30)	149.80 (83.73)	<0.001
Isobutyric acid, *mean (SD)*	431.10 (298.40)	469.10 (254.50)	0.744	501.50 (551.30)	312.30 (131.70)	0.607	304.20 (134.60)	293.20 (186.70)	0.882
Propionic acid, *mean (SD)*	40.22 (22.28)	42.78 (12.81)	0.251	49.59 (35.99)	27.30 (6.58)	0.210	13.77 (14.64)	5.59 (11.19)	0.385

C = control group without any treatment; L = probiotic group with *Lactobacillus acidophilus* D2/CSL; SD = standard deviation.

## Data Availability

The datasets analysed in the present study are available from the corresponding author on reasonable request.

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
