# Peer review of "Effects of Dietary Supplementation of Lactobacillus acidophilus on Blood Parameters and Gut Health of Rabbits"

_animals, 2022, doi:10.3390/ani12243543_

Round 1

Reviewer 1 Report

The article no -207944 by Colombino  et al entitled “Effects of dietary supplementation of Lactobacillus acidophilus D2/CSL on blood parameters and gut health of rabbits”. This article studied the effects of Lactobacillus acidophilus D2/CSL on biochemical traits, fecal score, cecal pH, gut histomorphometry, microbiota composition and fecal short-chain fatty acids in rabbits. The dietary inclusion of 1 x 109 cfu/kg feed once a day of Lactobacillus acidophilus D2/CSL did not impair rabbit productive traits, biochemical parameters, fecal score, gut morphometry, cecal pH, microbiota and SCFAs. However, it decreased health status and animal’s death, indicating that it could enhance disease resistance of rabbit. This article is very important and discuss good  issue related to GIT diseases are one of the most common reasons of rabbits’ death. Maintaining a proper gut health is fundamental to guarantee adequate growth performance and welfare of the animals. Probiotics (e.g., Lactobacillus acidophilus) have been proposed as good alternatives to positively modulate gut health. However, I have some comments that may improve the outcomes of this article:

1.   The novelty and /or the add value of good article should be focused in the introduction section.

2.    These references could help improve the literature background of the Ms.

- Shehata, A.A.; Yalçın, S.; Latorre, J.D.; Basiouni, S.; Attia, Y.A.; El-Wahab, A.A.; Visscher, C.; El-Seedi, H.R.; Huber, C.; Hafez, M. H; et al. Probiotics, Prebiotics, and
Phytogenic Substances for Optimizing Gut Health in Poultry. Microorganisms 2022, 10 (2), 395; https://doi.org/10.3390/microorganisms10020395

-Attia, Y.A.,  A. E. Abd El Hamid, A.M. Ismaiel and Maria C de  Oliveira, Mohammed A Al-Harthi, Asmaa Sh. El- Naggar  and G. A Simon (2018). Nitrate detoxification using antioxidants and probiotics in the water of rabbits. Revista Colombiana De Ciencias Pecuarias (RCCP),        2018; 31(2):130-138.

3.   L 137, Plz add form of the diet

4.     L 148-159,  Plz indicate the experimental unit for each trait, However, it is mentioned in L 239

5.     Plz connect the improvement in rabbits BCS and immune status with the productive traits of rabbits.

6.   The use of numbers and decimals in the tables should be follow the rule of: xxxx, xxx, xx.x, x.xx, 0.xxx and 0.0xxx

7.   References section References can be updated with 2022 so far publication

Author Response

The article no -207944 by Colombino  et al entitled “Effects of dietary supplementation of Lactobacillus acidophilus D2/CSL on blood parameters and gut health of rabbits”. This article studied the effects of Lactobacillus acidophilus D2/CSL on biochemical traits, fecal score, cecal pH, gut histomorphometry, microbiota composition and fecal short-chain fatty acids in rabbits. The dietary inclusion of 1 x 109 cfu/kg feed once a day of Lactobacillus acidophilus D2/CSL did not impair rabbit productive traits, biochemical parameters, fecal score, gut morphometry, cecal pH, microbiota and SCFAs. However, it decreased health status and animal’s death, indicating that it could enhance disease resistance of rabbit. This article is very important and discuss good  issue related to GIT diseases are one of the most common reasons of rabbits’ death. Maintaining a proper gut health is fundamental to guarantee adequate growth performance and welfare of the animals. Probiotics (e.g., Lactobacillus acidophilus) have been proposed as good alternatives to positively modulate gut health. However, I have some comments that may improve the outcomes of this article:

  1. The novelty and /or the add value of good article should be focused in the introduction section.

We added more information about the impact of L.acidophilus on gut health of rabbits, in order to make the focus on a novel strain more evident.

  1. These references could help improve the literature background of the Ms.

- Shehata, A.A.; Yalçın, S.; Latorre, J.D.; Basiouni, S.; Attia, Y.A.; El-Wahab, A.A.; Visscher, C.; El-Seedi, H.R.; Huber, C.; Hafez, M. H; et al. Probiotics, Prebiotics, and

Phytogenic Substances for Optimizing Gut Health in Poultry. Microorganisms 2022, 10 (2), 395; https://doi.org/10.3390/microorganisms10020395

-Attia, Y.A.,  A. E. Abd El Hamid, A.M. Ismaiel and Maria C de  Oliveira, Mohammed A Al-Harthi, Asmaa Sh. El- Naggar  and G. A Simon (2018). Nitrate detoxification using antioxidants and probiotics in the water of rabbits. Revista Colombiana De Ciencias Pecuarias (RCCP),        2018; 31(2):130-138.

According to reviewer’s suggestion, these references have been added in the manuscript (references number 10 and 18).

  1. L 137, Plz add form of the diet

We added the information.

  1. L 148-159, Plz indicate the experimental unit for each trait, However, it is mentioned in L 239.

The experimental unit is always the individual rabbit. This information is reported in the “Statistical analysis” subsection.

  1. Plz connect the improvement in rabbits BCS and immune status with the productive traits of rabbits.

We added a specific sentence to better explain these relationships.

  1. The use of numbers and decimals in the tables should follow the rule of: xxxx, xxx, xx.x, x.xx, 0.xxx and 0.0xxx

The authors are grateful to the reviewer for his/her comment. The format of number and decimals in the tables has been standardized as follow: two decimals for mean/median and standard deviation/interquartile range and three decimals for p-values. This is in accordance with the journal guidelines.

  1. References section References can be updated with 2022 so far publication.

According to the reviewer’s suggestion, the references section has been uploaded with the relevant 2022 publications.

Reviewer 2 Report

Elena Colombino and colleagues evaluated the effects of Lactobacillus acidophilus D2/CSL on biochemical parameters, fecal score, cecal pH, gut morphometry, microbiota composition and fecal short-chain fatty acids in rabbits. This manuscript is well written and the information provided is comprehensive. I only have a few comments that need to be addressed.

Line 205: what is the purpose to collect caecal content instead of other parts of intestine (e.g., ileal content)?

Line 212: which Qiime2 version did you use to analyze the data?

Line 269: For the P value in this manuscript, do we have to make the “P” in italics? Please double check with the journal.

Line 307-308: How do you get the conclusion that non-significant differences were observed between C and L group for beta diversity? Did you run the ANOSIM or PERMANOVA analysis to confirm that there are no significant different between these two groups? Also, where is the Alpha results?

Line 315: for the figure 1b, please enlarge the legend for the figure. Also, please specify which taxonomy level that you are presenting, is it family or genus level?

Line 322: dietary supplementation of Lactobacillus acidophilus D2/CSL did not affect acetic acid, isobutyric acid and propionic acid concentration, so the P value should be <0.05 instead of  > 0.05.

Line 327: In the L group, the rabbits were treated with Lactobacillus acidophilus, how is the relative abundance of this strain in L group compared with C group? Is it increased or decreased? It might be hard to detect the relative abundance of L. acidophilus in these two groups since V3-V4 region is too short to detect the bacteria in species level, but we may can provide the relative abundance of Lactobacillus (genus level), it also can provide some valuable information for us.

Also, lefse analysis (or similar analysis) should be applied to detect the different bacteria between these two groups (at genus, ASV level, or any taxonomy level that you are interested in).

Author Response

Elena Colombino and colleagues evaluated the effects of Lactobacillus acidophilus D2/CSL on biochemical parameters, fecal score, cecal pH, gut morphometry, microbiota composition and fecal short-chain fatty acids in rabbits. This manuscript is well written and the information provided is comprehensive. I only have a few comments that need to be addressed.

Line 205: what is the purpose to collect caecal content instead of other parts of intestine (e.g., ileal content)?

In rabbits, cecum is considered the main fermenter organ. For this reason, most studies that aimed to unravel rabbit’s intestinal microbiota have been focused on the characterization of cecal microbial communities (Velasco-Galilea et al., 2018; Front. Microbiol., 13 September 2018 Sec. Microbial Symbioses https://doi.org/10.3389/fmicb.2018.02144 Rabbit Microbiota Changes Throughout the Intestinal Tract).

Line 212: which Qiime2 version did you use to analyze the data?

QIIME 2 v. 2022.8 the information was added to the revised manuscript

Line 269: For the P value in this manuscript, do we have to make the “P” in italics? Please double check with the journal.

The authors are thankful to the reviewer for his/her comment. All the P value have been written in italics in the manuscript.

Line 307-308: How do you get the conclusion that non-significant differences were observed between C and L group for beta diversity? Did you run the ANOSIM or PERMANOVA analysis to confirm that there are no significant different between these two groups? Also, where is the Alpha results?

Regarding microbiota analysis, alpha and beta diversity indexes were calculated through the diversity script of QIIME2. Differences between alpha and beta diversity parameters were calculated with Kruskal-Wallis and ANOSIM statistical test respectively. and ASVs frequency at the lowest taxonomic resolutions (genus or family) were analysed by non-parametric Kruskall-Wallis test in R environment. Figure 1 was modified according to reviewer suggestion to display alpha and beta diversity calculations.

Line 315: for the figure 1b, please enlarge the legend for the figure. Also, please specify which taxonomy level that you are presenting, is it family or genus level?

Corrected as requested.

Line 322: dietary supplementation of Lactobacillus acidophilus D2/CSL did not affect acetic acid, isobutyric acid and propionic acid concentration, so the P value should be <0.05 instead of  > 0.05.

Corrected as requested.

Line 327: In the L group, the rabbits were treated with Lactobacillus acidophilus, how is the relative abundance of this strain in L group compared with C group? Is it increased or decreased? It might be hard to detect the relative abundance of L. acidophilus in these two groups since V3-V4 region is too short to detect the bacteria in species level, but we may can provide the relative abundance of Lactobacillus (genus level), it also can provide some valuable information for us. Also, lefse analysis (or similar analysis) should be applied to detect the different bacteria between these two groups (at genus, ASV level, or any taxonomy level that you are interested in).

As pointed out from the reviewer the metataxonomic approach was not able to discriminate Lactic acid bacteria due to the short size of the amplicons. In addition, the frequency of LAB was very low due the highest background of the cecal microbiota. This is not surprising since several paper using the same approach in study based on LAB administration did not find LAB in the gut due to the highest background of resident microbiota.